# Assessment of Seismic Bedrock in Deep Alluvial Plains. Case Studies from the Emilia-Romagna Plain

Luca Martelli 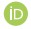

Geological, Seismic and Soil Survey, Emilia-Romagna Region, 40127 Bologna, Italy;
luca.martelli@regione.emilia-romagna.it

**Abstract:** The estimation of seismic shaking is essential for a realistic assessment of the local seismic hazard and the implementation of effective strategies for prevention and mitigation of the seismic risk. One of the most important aspects in the analysis of the site seismic assessment is the recognition of the seismic bedrock and its depth. Unfortunately, these data are not always easy to evaluate, especially in areas where the thickness of loose or poorly consolidated sediments is high. This article illustrates data and case studies from the Emilia-Romagna sector of the Po Plain, in order to provide examples and suggestions for the recognition of the seismic bedrock in alluvial and coastal areas characterised by significant thicknesses of unconsolidated sediments, using available data and not expensive geophysical surveys. The application of the proposed method indicates that the study area can be divided into four domains characterized by different depths of the seismic bedrock: the marginal or pede-Apennine belt, the high structural zones, the syncline/minor anticline zones, and the Po delta-coast zone.

**Keywords:** seismic bedrock; geological substratum; ambient vibrations; Vs propagation; Po Plain stratigraphy

## 1. Introduction

It is known that loose and poorly consolidated sediments, such as recent alluvial deposits or slope debris, can change the amplitude, frequency, and duration of the seismic motion (e.g., [1,2]).

Therefore, it is very important for the evaluation of the expected seismic motion to know the thickness of loose sediments, poorly consolidated deposits and weathered or fractured rocks, i.e., to identify the depth below which the lithotypes have rigid behaviour.

The recognition of the rigid substratum, or seismic bedrock, can be complex in the case of non-rigid sediments with a thickness greater than 50 m, considering that the economic resources usually available only allow the use of current geophysical techniques (down-hole, MASW, ReMi), which rarely allow investigation beyond this depth.

This problem arises both in seismic microzonation studies for urban planning and in the estimates of expected seismic motion for design, and therefore has significant implications in the prevention and reduction of seismic risk.

Mascandola et al. (2019) [3] already discussed this topic and proposed a map of the seismic bedrock in the Po Plain, based on environmental noise measurements, from pre-existing and new data. The article is undoubtedly a fundamental reference on the subject, but the scale of the map does not allow its direct application, neither for urban planning, which operates at a municipal or inhabited centre scale, nor at the site scale for the design. Furthermore, some in-depth investigations carried out for seismic microzonation studies, have shown that the surface mapped by Mascandola et al. (2019) [3] does not always correspond to the roof of the engineering seismic bedrock, i.e., the surface below which the shear wave velocity (Vs) reaches and exceeds 800 m/s. Finally, the study by Mascandola et al. (2019) [3] has not been extended to the southern plain, near the Apennines hills.

The aim of this article is to provide useful information for the assessment of the seismic bedrock depth in wide and deep alluvial basins, such as the Po Plain, using data available in the literature and/or acquired through relatively inexpensive surveys. For this purpose, case studies in significant sites of the Emilia-Romagna plain will be illustrated.

The proposed procedure is based on the comparison between Vs profiles, stratigraphic data and values of the resonance frequency of the ground (Fn) acquired with recordings of environmental vibrations [4].

It is commonly accepted in the literature (e.g., [5,6]) that Fn depends on the thickness of the unconsolidated sediments (H) and the average speed of the shear waves in the thickness range H ($\overline{V}_S$) according to the relationship:

$$F_n = \frac{\overline{V}_S}{4H} \tag{1}$$

It is therefore clear how, by having Fn values, Vs profiles and information on the subsoil lithostratigraphy, it is possible to estimate with a certain reliability the thickness H, i.e., the depth of the main resonant surface comparable to the roof of the bedrock.

Seismic microzonation studies carried out in the Emilia-Romagna plain (https://geo.regione.emilia-romagna.it/schede/pnsrs/, accessed on 1 July 2021) indicate that the interest range of the ground frequency Fn is between 0.2 Hz and 20 Hz and that, in general terms, thickness ranges of unconsolidated sediments can be associated with Fn values (Table 1). The overlaps of the thickness ranges in Table 1 depend on the Vs variability of the unconsolidated sediments.

**Table 1.** Comparison between ground frequency and unconsolidated sediments thickness in the Emilia-Romagna plain (modified from [7]).

| Ground Frequency | Unconsolidated Sediments Thickness |
|:---:|:---:|
| Fn $\leq$ 0.6 Hz | H $\geq$ 130 m |
| 0.6 < Fn $\leq$ 1.2 Hz | 170 $\leq$ H < 50 m |
| 1.2 < Fn $\leq$ 2 Hz | 80 $\leq$ H < 20 m |
| 2 < Fn $\leq$ 5 Hz | 30 $\leq$ H < 8 m |
| 5 < Fn $\leq$ 20 Hz | H $\leq$ 12 m |

The use of ambient vibrations for the seismic-stratigraphic characterization of the subsoil is not new (e.g., [5]), but even today, the application of this procedure is not always optimal as the interpretation is often not supported by the comparison with stratigraphic data, even though these are readily available.

The stratigraphic data are of great importance especially in the case of tectonically active basins, such as the Po Plain. In this kind of basins, the sedimentation is affected not only by glacio-eustatic variations, but also by the tectonic phases with the result that the stratigraphic succession is characterized by sedimentary cycles of various hierarchical order to which correspond stratigraphic units bounded by discontinuity surfaces. The level of deformation decreases from the oldest to the most recent stratigraphic units. This tectono-stratigraphic evolution confers a different degree of compaction and stiffness of the sediments, which decreases towards the top, and the surfaces of stratigraphic discontinuity that separate the various units are also surfaces of discontinuity of stiffness.

*Definitions of Cover Sediments, Geological Substratum, Seismic Bedrock*

Geophysical investigations, conducted both in mountain areas and in the plains, show that the lithotypes characterised by rigid behaviour do not always coincide with the geological substratum of the cover sediments.

To understand this difference, it is important to define "cover sediments", "geological substratum", and "seismic bedrock".

Cover sediments: stratigraphic succession consisting of loose and/or poorly consolidated sediments, such as slope debris, recent alluvial deposits (e.g., younger than 1 Ma), and/or weathered/fractured rocks. Generally, these sediments are characterised by Vs < 400 m/s and assume importance for the site seismic assessment if the thickness is greater than 3 m.

Geological substratum: stratigraphic succession generally made up of diagenised or consolidated rocks, at the base of the cover sediments. In the Po Plain, the geological substratum of the alluvial sediments is made up of consolidated marine sediments.

Seismic bedrock: stratigraphic succession characterised by rigid behaviour, that is rocks characterised by Vs significantly higher than that of the sediments above (e.g., $Vs_{bedrock}/Vs_{cover} > 2$). In the design rules (e.g., [8,9]), the seismic bedrock is defined as rock or rock-like geological formation characterised by Vs > 800 m/s (ground type A); for this reason, a stiff geological unit characterised by $Vs \geq 800$ m/s is also indicated as "engineering bedrock" (e.g., [3]).

To better understand the importance of the above definitions, it is important to consider that the geological substratum can act as a seismic bedrock even in the case of Vs < 800 m/s, if the seismic impedance contrast between substratum and cover is significant.

Figure 1 shows geophysical tests carried out on a landslide in the Northern Apennines [10]. In this site, ambient noise measurements show a peak of the horizontal to vertical spectral ratios (H/V) at the frequency of 3.6–3.8 Hz. To identify the depth of the surface responsible for this peak, a borehole was drilled, and a direct Vs measurement (down-hole) was carried out. Landslide debris was drilled up to a depth of 25 m and then a substratum consisting of compact grey marls was drilled up to the bottom hole (40 m). The grey marls can be correlated with the Montepiano Marls (upper Eocene–Lower Oligocene), extensively outcropping in the surrounding areas. The down-hole test recorded 300 < Vs < 450 m/s in the landslide debris and 600 < Vs < 750 m/s in the first 13 m of substratum. The results of the two tests are perfectly consistent, and this made it possible to recognise that the Montepiano Marls behave like seismic bedrock, even though Vs < 800 m/s (in the tested thickness).

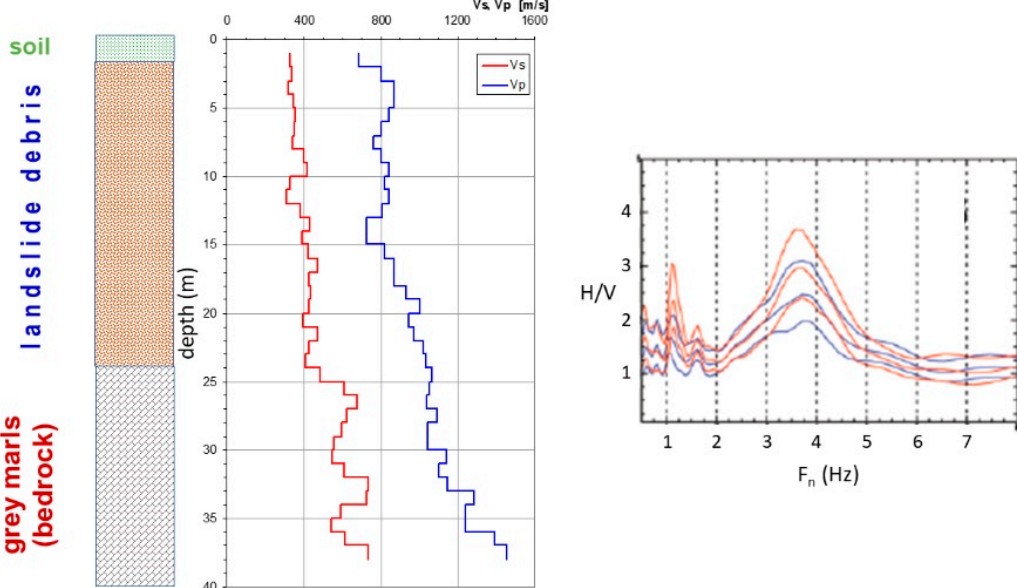

**Figure 1.** Comparison between stratigraphic log, Vs profiles from down-hole and ambient noise in an Apennines site characterised by debris cover (landslide) on marly geological substratum (modified from [10]).

Similar cases are frequent in the Apennines.

## 2. Useful Documents

The first step in the recognition of the seismic bedrock is to reconstruct the geological model. The deep geological cross-sections are essential in order to identify the geological substratum.

The sheets of the Geological Map of Italy 1: 50,000, and related sketches, explanatory notes, and cross-sections, are available on the website of the Geological Survey of Italy: https://www.isprambiente.gov.it/Media/carg/ (accessed on 1 July 2021). Some sheets of plain areas are accompanied by subsoil maps, which represent the main buried geological elements (roof of the sandy/gravelly horizons, unconformities, tectonic structures, roof of the geological substratum, ... ).

As regards the first 500–600 m of subsoil of the Po Plain valuable information is available in the studies carried out by Emilia-Romagna Region and Lombardy Region, in collaboration with ENI, for the characterisation of the water resources [11,12]. In these publications, maps of the isobaths and thicknesses of the main aquifers and cross-sections up to the geological substratum are available. A similar study, although with less detailed maps and geological cross-sections, was also carried out by the Piedmont Region, in collaboration with the CNR and the University of Turin [13].

The study on the aquifers of the Ferrara Province is a local update on the Po Plain subsoil [14].

The most important geological cross-sections (carried out by the regional Geological, Seismic and Soil Survey) and some geotechnical tests in the Emilia-Romagna plain are available on the Emilia-Romagna Region website https://geo.regione.emilia-romagna.it/cartografia_sgss/user/viewer.jsp?service=sezioni_geo (accessed on 1 July 2021).

Finally, useful information for the geological, geotechnical, and geophysical characterisation of the subsoil can be found in seismic microzonation studies. Italian seismic microzonation studies, carried out with the funds of Law 77/2009, art. 11, are available on the DPC website https://www.webms.it/ (accessed on 1 July 2021), managed by CNR-IGAG. Seismic microzonation studies of the Emilia-Romagna Region are also available on the website http://geo.regione.emilia-romagna.it/schede/pnsrs/ (accessed on 1 July 2021).

## 3. Geological Framework

The Po Plain is the largest alluvial plain in Italy and is limited by the Southern Alps to the north, the Western Alps to the west, the Northern Apennines to the south and the Adriatic Sea to the east (Figure 2). The Emilia-Romagna plain constitutes the central-southern and eastern sectors.

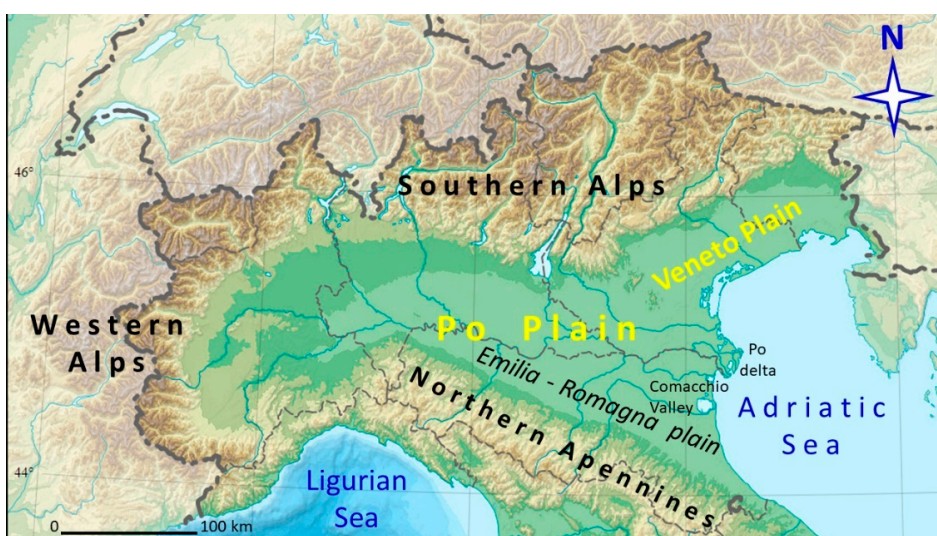

**Figure 2.** Geographical framework of the Po Plain.

The Po Plain represents the foredeep of two chains with opposite vergence. The Northern Apennines, south side, are north-verging, and their front does not coincide with the morphological transition from the hill to the plain but is made up of buried thrusts (Figure 3) belonging to the Emilia Folds and Ferrara Folds [15]. The Southern Alps, north side, have vergence towards the south and the front is also buried by the alluvial deposits. Therefore, the subsoil of the Po Plain is structured into anticlines and synclines, and the thickness of alluvial sediments is very variable, even in the distal areas: from over 500–600 m in the synclines up to a few tens of meters on the anticlines (Figure 4).

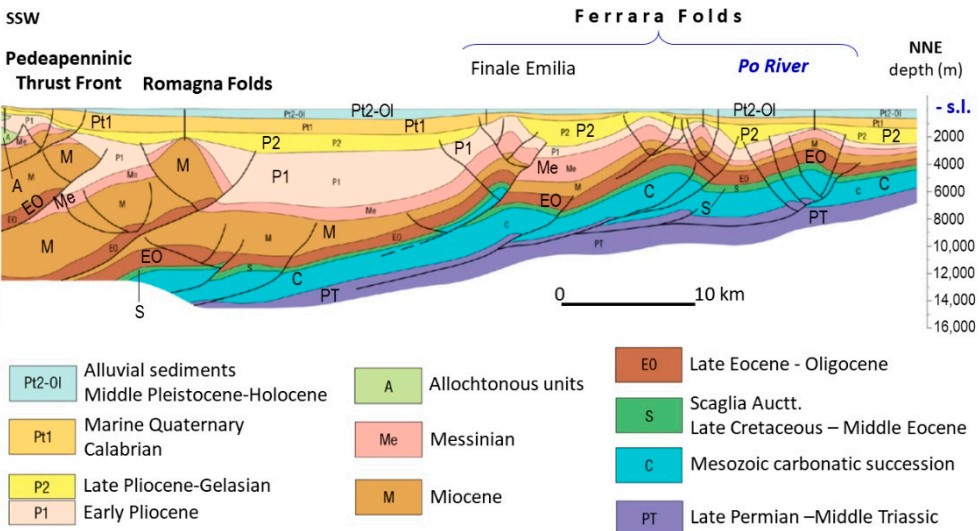

**Figure 3.** Geological cross-section from the Bologna Apennines to the Po River [16]; trace in Figure 4.

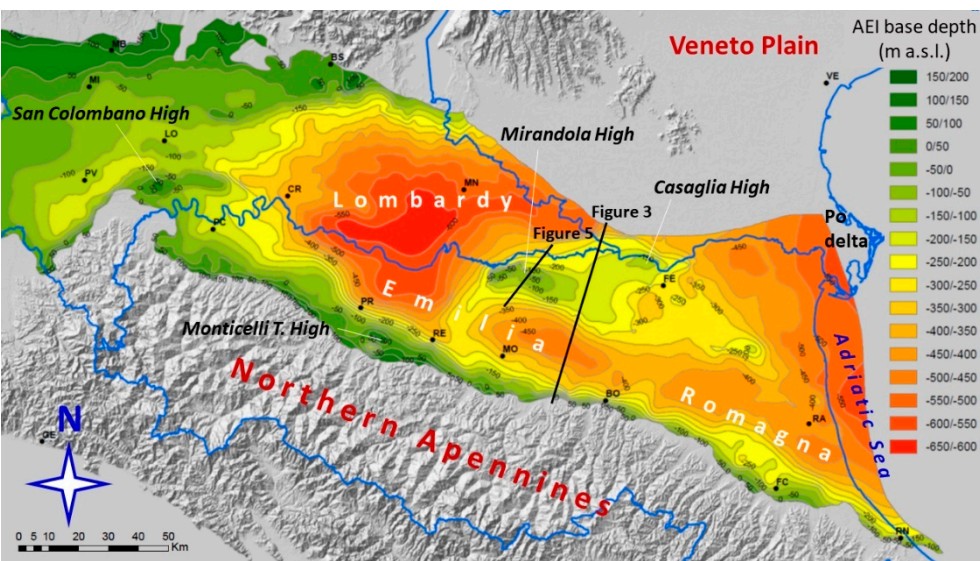

**Figure 4.** Map of the basal unconformity of the alluvial deposits in the Lombard and Emilia-Romagna plain (isobaths referred to the sea level) (modified from [11,12]); the black dots indicate the main cities.

The alluvial sediments of the Emilia-Romagna plain are the result of the depositional activity of the Po River and its right tributaries and can be divided into two main stratigraphic units (Figure 5): the lower Emilia-Romagna Synthem (0.65–0.45 Ma) and the Upper Emilia-Romagna Synthem (0.45 Ma–Present), both limited at the base by discontinuity surfaces [11,12]. The literature indicates these Synthems, respectively named "Sintema Emiliano-Romagnolo Inferiore" and "Sintema Emiliano-Romagnolo Superiore" in Italian, with the acronyms AEI and AES (see Geological Map of Italy:

https://www.isprambiente.gov.it/Media/carg/emilia.html, accessed on 1 July 2021) or SERI and SERS [16–18]. Both Synthems can be divided into subunits made up of alternations of coarse (sands and gravels) and fine (clays and silts) sediments. The coarse sediments are obviously prevalent in the marginal areas while the successions of the distal areas, both in the synclines and on the anticlines, are made up of prevailing fine sediments (clays and silts), with intercalations of sandy horizons. The most superficial deposits, Holocene in age, are loose sediments; the consolidation degree of the underlying sediments increases with depth; however, the alluvial sediments of the Po Plain, even those at greater depth, are poorly consolidated and susceptible to subsidence.

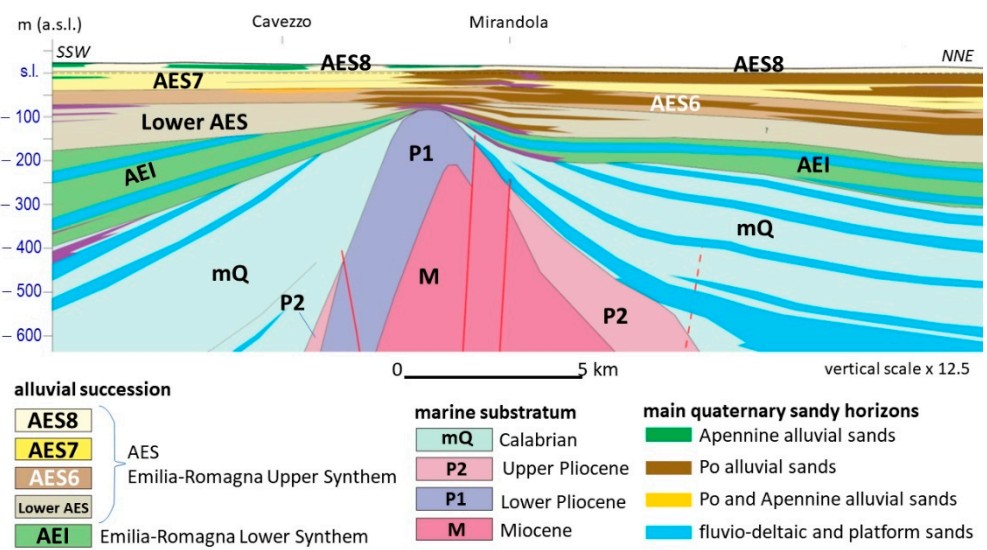

**Figure 5.** Geological cross-section through the Modena plain and the Mantua Province south of the Po River [19]; trace in Figure 4.

The substratum is generally made up of transitional sands, Middle Pleistocene in age (Imola Sands, Yellow Sands, Costamezzana Synthem *Auctt.*), and marine sediments (alternating clay, marl, and sandstone), Pliocene–Lower Pleistocene in age (known as Argille Azzurre *Auctt.*), extensively outcropping in the Apennines hills [11,20].

Apennines and Alps are still-forming chains, as evidenced by the frequent seismicity recorded on the front of the two chains (e.g., [18,21]). Therefore, the tectonic activity influenced the sedimentation of all geological units, whose basal discontinuity surfaces are characterised by angular unconformity, more evident at the basin margins and in the high structural areas (Figure 5).

## 4. Case Studies

Based on the variability of the thickness of the alluvial cover and the distribution of lithotypes, large alluvial basins can be divided into:

(1) Marginal areas;

(2) Distal areas, which, if tectonically articulated, can in turn be divided into high structural areas (top of the anticlines) and other areas (limbs and synclines).

The Emilia-Romagna plain includes all areas indicated above. Thanks to the numerous studies of the site seismic assessment carried out in the last 10 years, it is possible to illustrate examples of seismic bedrock determination in each of these areas.

In particular, the examples consider data and results from studies carried out for the characterisation of the environmental effects observed during the Emilia 2012 earthquake [3,22–26] and the seismic microzonation for the reconstruction [19] and for the urban planning (see https://geo.regione.emilia-romagna.it/schede/pnsrs/, accessed on 1 July 2021), as well as important studies for the evaluation of stability in seismic conditions of major works [27–31].

Figure 6 shows the locations of the case studies illustrated in this paper.

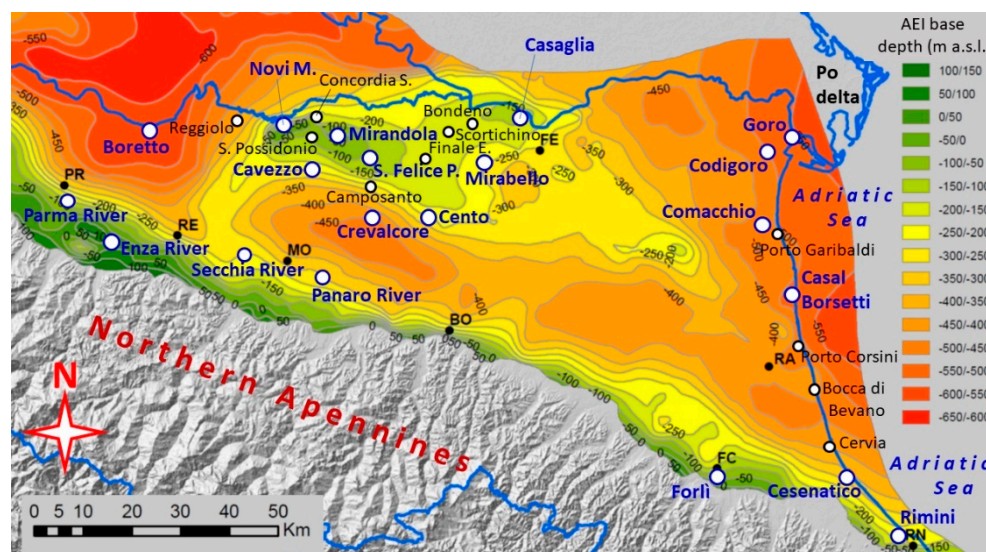

**Figure 6.** Location of the case studies (white dots). Larger symbols and blue toponyms indicate sites with geophysical/geotechnical tests and geological cross-sections; smaller symbols and black toponyms indicate sites with only geophysical tests.

The most significant examples are illustrated below. Other examples can be found in the Supplementary Materials.

*4.1. Marginal Area*

The marginal area of the Emilia-Romagna plain is the southern zone, close to the Northern Apennines. The southern coast, between Rimini and the Gabicce promontory (Marche region), is also part of this area.

This sector is characterised by prevalently coarse sequences, made up of plurimetric gravelly horizons, generally amalgamated towards the mountain, where the thicknesses of gravels can also be decametric, and interbedded with sandy and silty horizons towards the plain. The geological substratum is usually made up of transitional sands of Middle Pleistocene age and/or marine clays and silts of Pliocene–Lower Pleistocene age.

Geophysical surveys indicate that, generally, in gravelly horizons, Vs is greater than 400 m/s while the Vs in the first meters of the geological substratum is about 300–400 m/s. Figure 7 shows an example of stratigraphic log and Vs profile (from down-hole test) in an alluvial fan of the Apennines-plain margin.

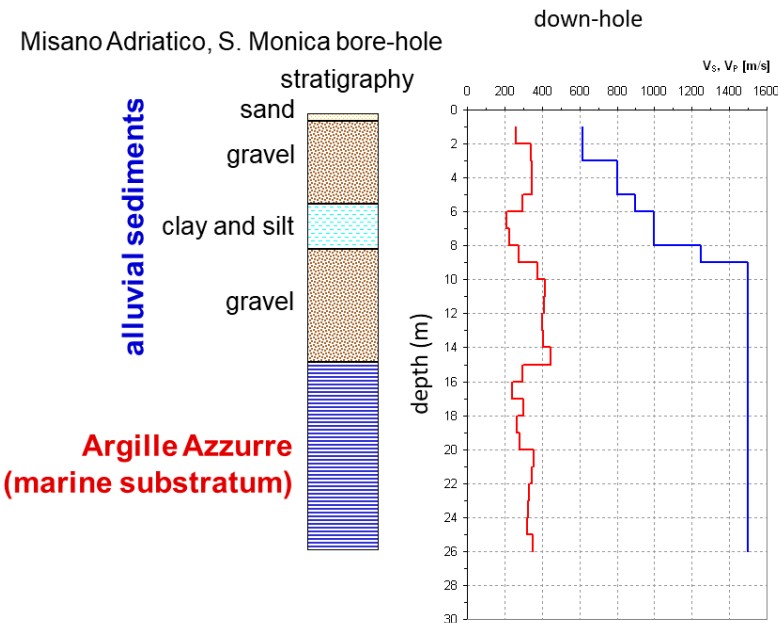

**Figure 7.** Example of stratigraphic log and Vs-Vp profiles (down-hole) in the foothills area (Santa Monica site, Misano Adriatico Municipality, Rimini Province).

Figure 8 shows a Vs profile (from a cross-hole test) in the southern area of the city of Forlì (from http://itaca.mi.ingv.it/ItacaNet_30/#/station/IT/FOR, accessed on 1 July 2021), where the subsoil is characterised by alluvial fan deposits (Montone, Rabbi and Bidente Rivers) and the geological substratum is deeper than 200 m. The figure shows that in the first gravelly horizon, between 12 and 15 m, Vs = 400 m/s and in the second gravelly horizon, below the depth of 42–43 m, Vs $\geq$ 800 m/s. The ambient noise measurement shows a peak of the H/V spectral ratio around 1.7–1.8 Hz, consistent with the Vs profile.

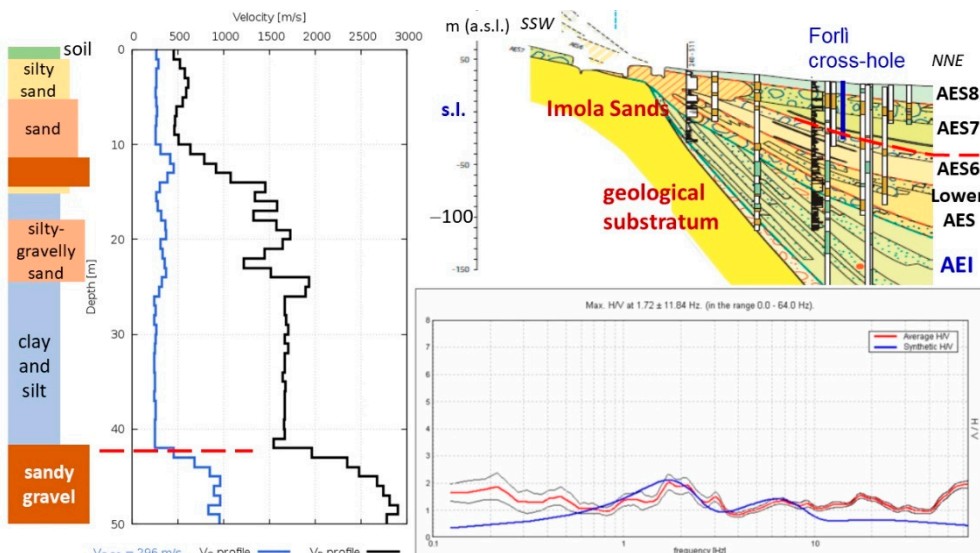

**Figure 8.** Stratigraphic log, Vs-Vp profiles (cross-hole, from http://itaca.mi.ingv.it/ItacaNet_30/#/station/IT/FOR, with modifications, accessed on 1 July 2021) and ambient noise measurement (from seismic microzonation, Forlì Municipality) in the Forlì station of the National Accelerometric Network; geological cross-section from CARG 1:50,000, sheet n. 240 "Ravenna".

The comparison between geophysical and stratigraphic data shows that the main Vs discontinuity corresponds to the stratigraphic discontinuity that separates the sub-synthems AES7 and AES6.

Similar information derives from the cross-hole test carried out in Viserba (Rimini Municipality), in a 96 meters deep borehole (Figure 9) in the alluvial fan of the Marecchia River [32,33], see also Figure S1 in the Supplementary Materials.

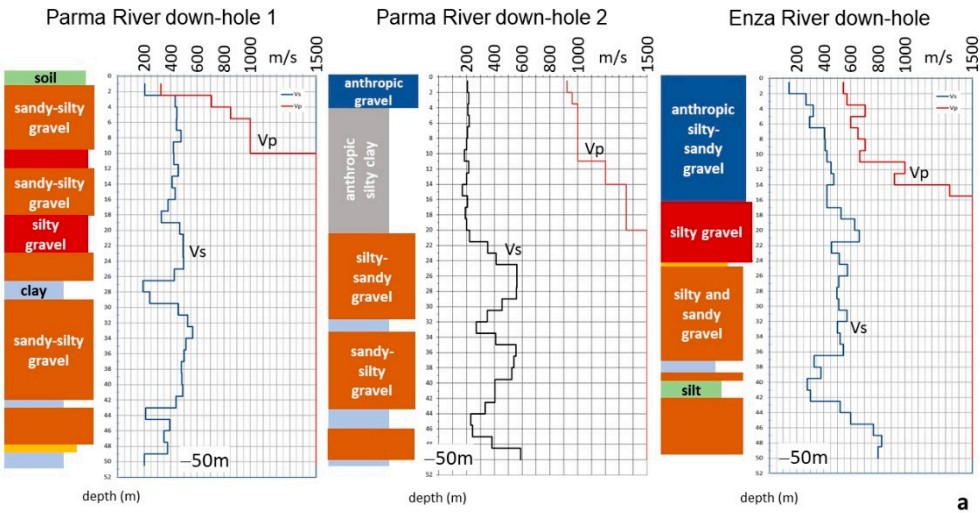

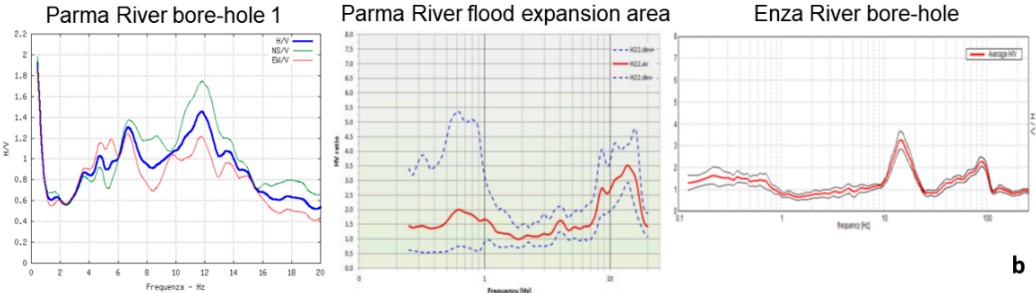

**Figure 9.** (**a**) Stratigraphic logs and Vs-Vp profiles ([27], with modifications) and (**b**) ambient noise measurements (from seismic microzonations, Parma Municipality and Montecchio Emilia Municipality) in the alluvial fan areas of the Parma and Enza Rivers.

High Vs values were also measured in the gravels of the alluvial fans of the Parma, Enza, Secchia and Panaro Rivers [27,29]. Down-hole tests provided Vs > 400 m/s already in the first gravelly horizons, at depths lower than 20 m, and higher Vs values, often greater than 600 m/s, in gravels at depths of 30–40 m (Figures 9 and 10).

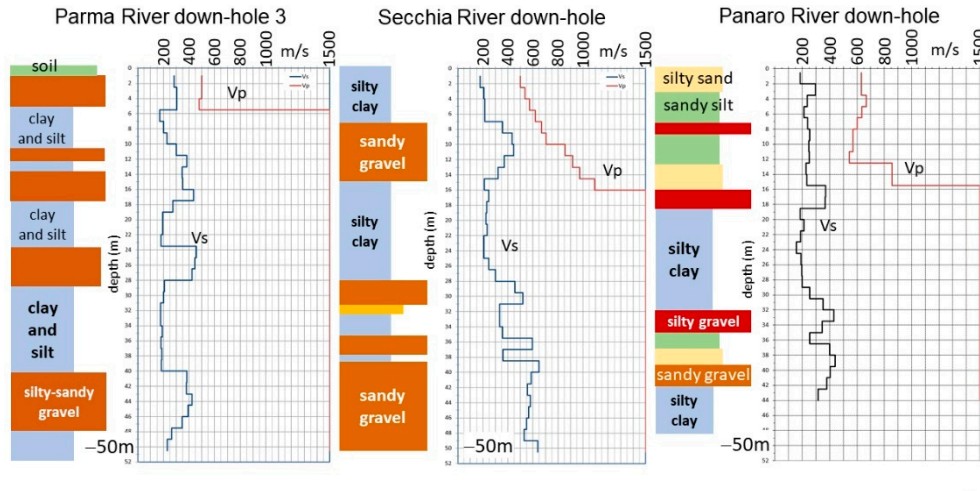

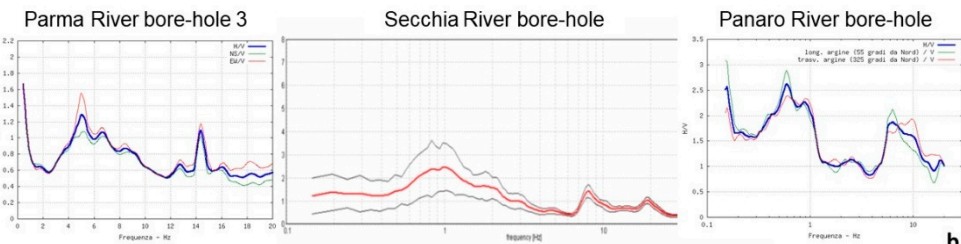

**Figure 10.** (**a**) Stratigraphic logs and Vs-Vp profiles ([27], with modifications) and (**b**) ambient noise measurements (from seismic microzonations, Parma Municipality, Campogalliano Municipality, and Modena Municipality) in the alluvial fan areas of the Parma, Secchia and Panaro Rivers.

The down-hole tests carried out in the Parma and Enza Rivers are both located on the top of a buried structural high (Monticelli Terme) where the thickness of alluvial sediments varies from about 60–70 m to 100–110 m (Figures 11 and 12), whereas the down-hole tests relating to the Secchia and Panaro Rivers are in areas with high thicknesses of alluvial sediments, (about 300 m, Figures 13 and 14).

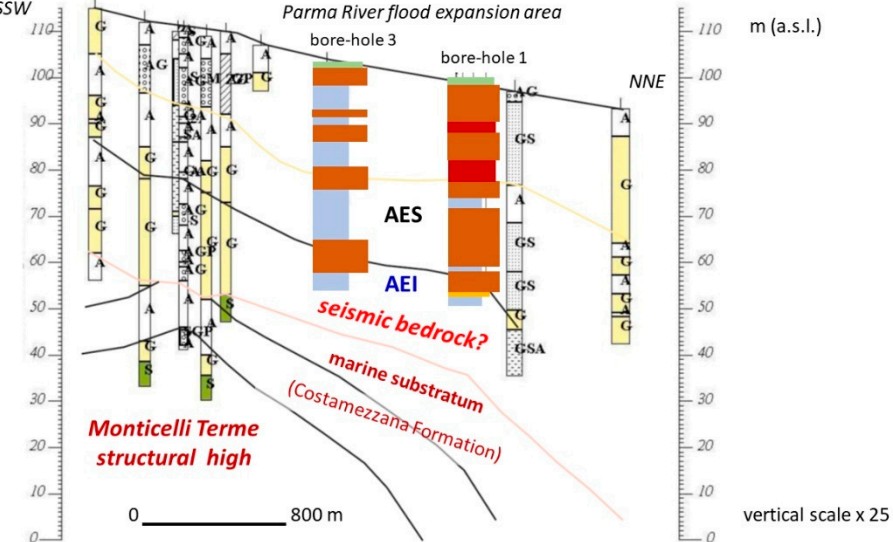

**Figure 11.** Geological cross-section across the southern side of the Parma River alluvial fan ([27], with modifications).

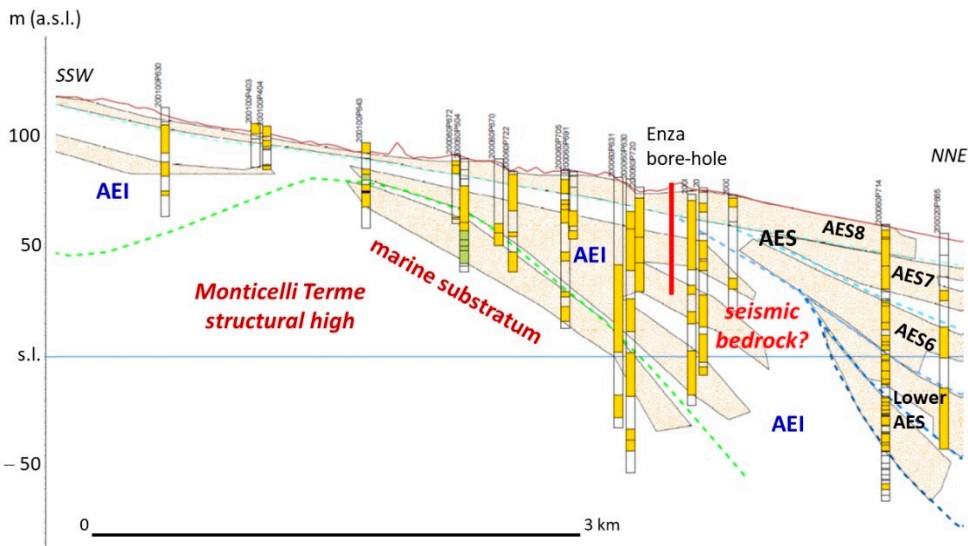

**Figure 12.** Geological cross-section across the Enza River alluvial fan (from https://geo.regione. emilia-romagna.it/cartografia_sgss/user/viewer.jsp?service=sezioni_geo, accessed on 1 July 2021, with modifications).

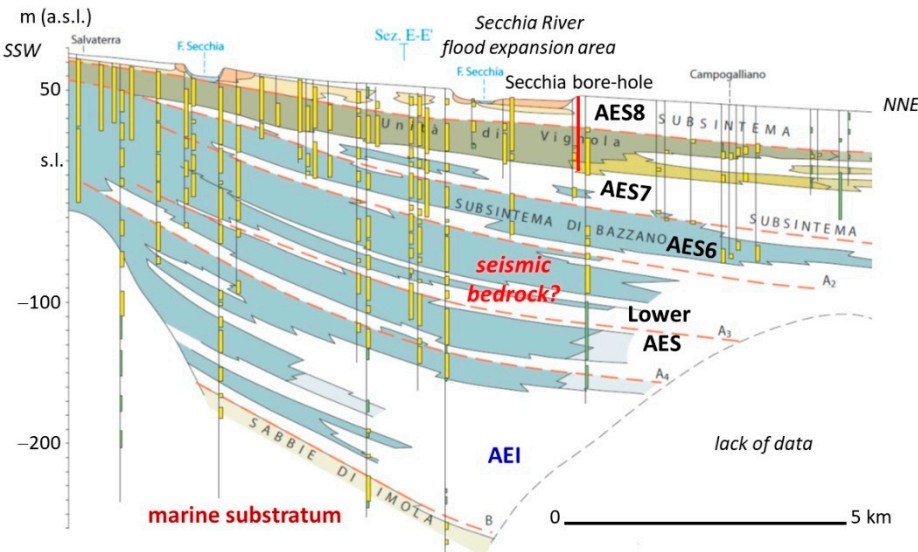

**Figure 13.** Geological cross-section across the Secchia River alluvial fan (from CARG 1:50,000, sheet n. 201 "Modena", with modifications).

The ambient noise measurements show a single high frequency peak (Fn > 10 Hz) where the succession is almost exclusively made up of gravels, indicating that the main Vs contrast surface is the roof of the first gravel horizon (Figures 9, 11 and 12). Where the succession consists of gravelly and silty-clayey horizons (Figures 10, 13 and 14) the ambient noise measurements show two peaks, the main one at lower frequencies and the secondary one at high frequencies (Fn > 5 Hz, sometimes Fn > 10 Hz), indicating that, in this case as well, the roof of the first gravelly horizon is an important impedance contrast surface, but the most important contrast surface of Vs is deeper.

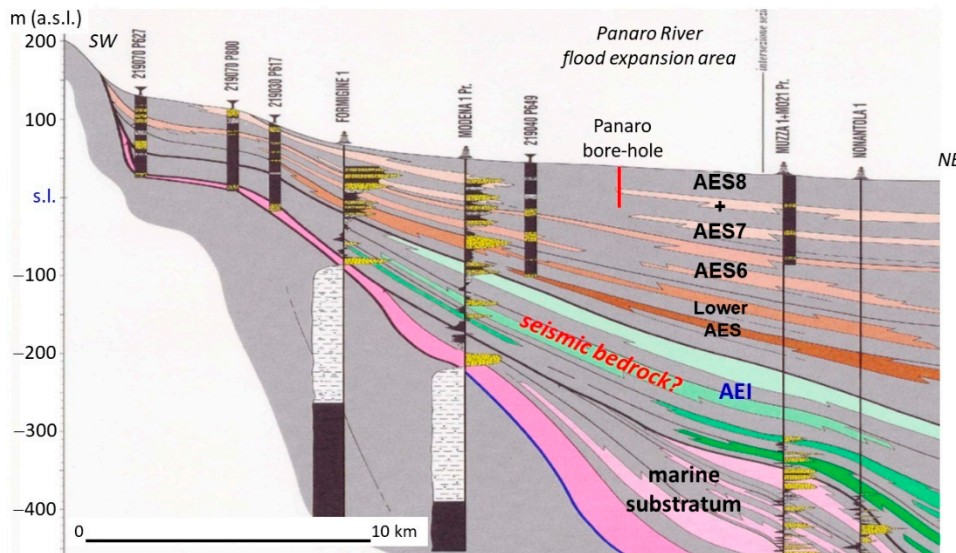

**Figure 14.** Geological cross-section across the Panaro River alluvial fan (modified from [11]).

*4.2. Distal Areas*

The geological substratum of alluvial sediments in the Po Plain is structured in anticlines and synclines due to the activity of blind thrusts. This determines a high variability in the thickness of alluvial sediments: at the top of the higher anticlines, it is reduced to a few tens of meters (San Colombano al Lambro, Monticelli Terme, Mirandola) while in the synclines it exceeds 600 m (south-eastern Lombardy plain) (Figure 4).

Measurement of ambient noise conducted along the Po River [28,30] and in the epicentral areas of the Emilia 2012 seismic sequence [19,22,23] highlighted two resonance frequencies of the ground: one at 0.8–0.9 Hz and the other at 0.25–0.3 Hz, with the exception of the buried structural highs (Mirandola and Casaglia) where a single ground frequency was detected, ranging from 0.8 Hz to 1.4 Hz, with a peak of the H/V spectral ratio usually greater than 3 (see also [34]).

Based on these observations, the description of the case studies in the Emilia-Romagna plain will be distinguished in high structural zones (top of the higher anticlines) and other areas (synclines, limbs, and lower anticlines).

4.2.1. High Structural Zones

In the Mirandola and Casaglia areas, located on the southern and northern buried ridges of the Ferrara Folds [16], boreholes were drilled up to the geological substratum of the alluvial deposits (Figure 15), made up of pre-Quaternary marine sediments (marly clays of Pliocene age in the Mirandola site, marls of Tortonian–Messinian age in the Casaglia site). In both bore-holes the Vs was measured with cross-hole tests and the results indicate that in the geological substratum the Vs is significantly higher than that of the alluvial sediments and exceeds 800 m/s [19,22].

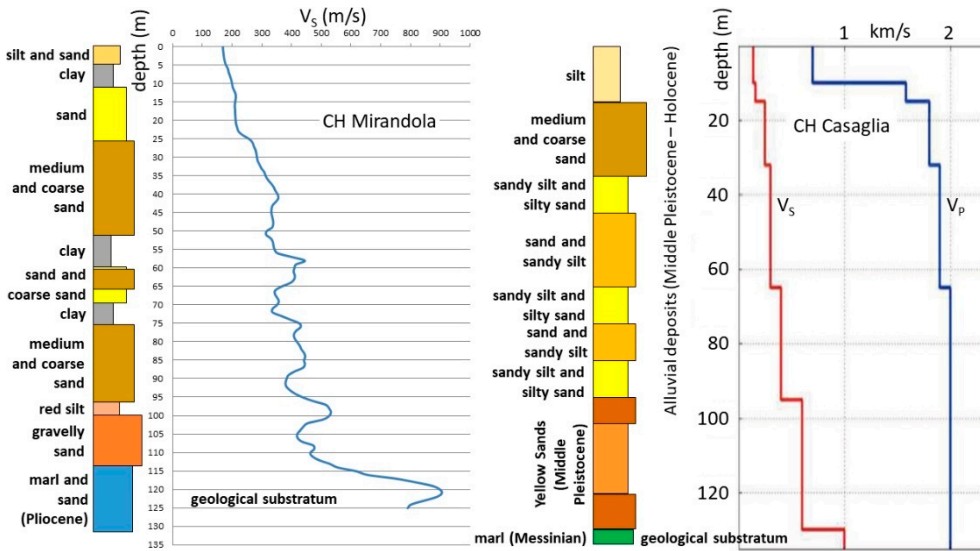

**Figure 15.** Stratigraphic logs and Vs profiles in Mirandola and Casaglia [19,22].

Results of ambient noise measurements carried out next to the boreholes (Figures 16 and 17) show, in both cases, a single peak of the H/V spectral ratio at a frequency of 0.9 Hz with high amplitude [19,22–24,34,35].

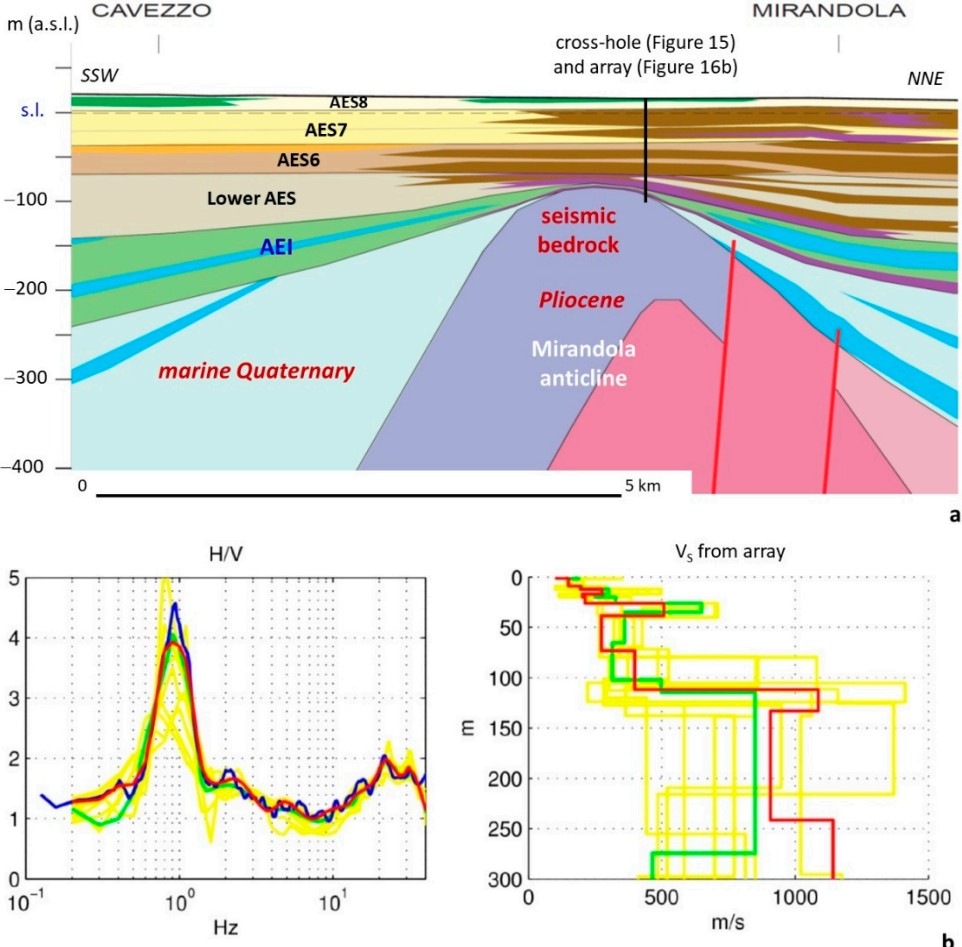

**Figure 16. (a)** Geological cross-section across the Mirandola anticline, **(b)** H/V spectral ratio and Vs profile from ambient vibration measurement (array) in the Mirandola urban area [19].

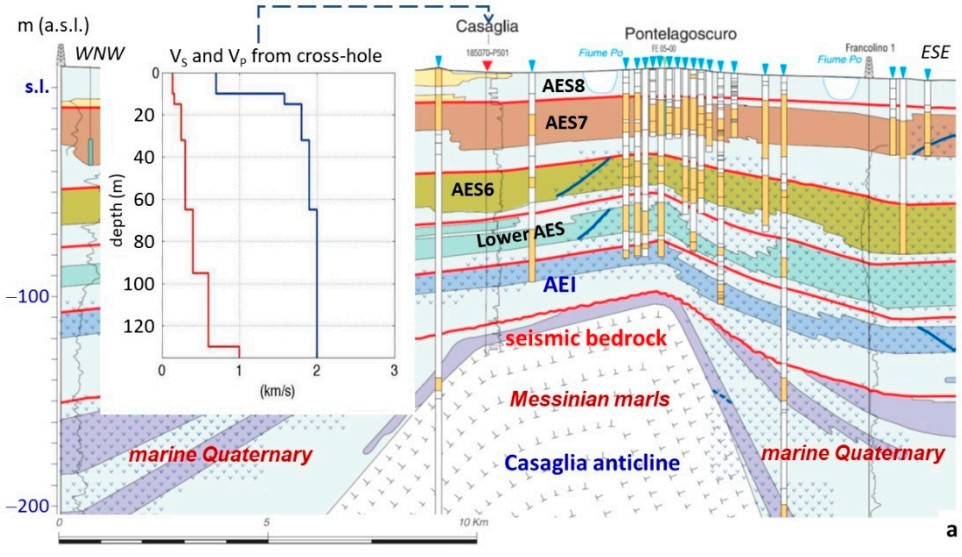

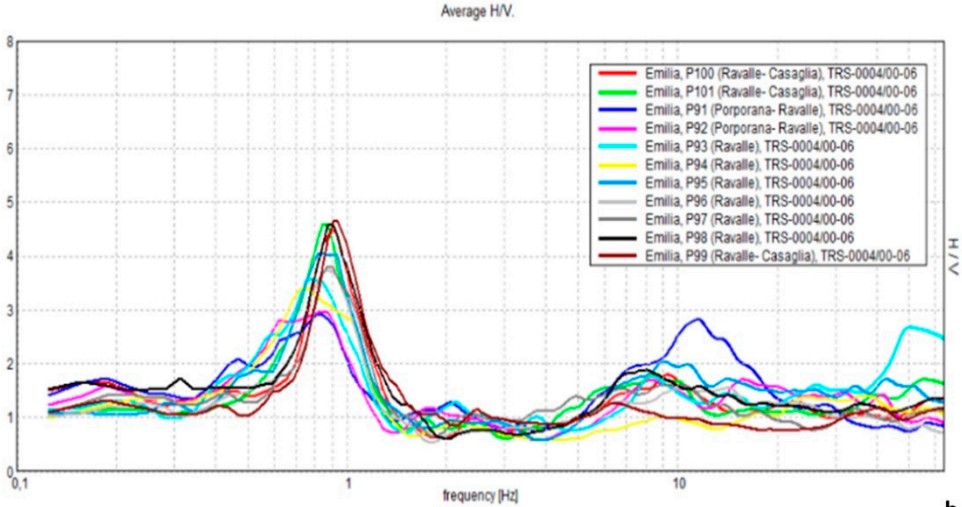

**Figure 17. (a)** Geological cross-section across the Casaglia anticline [14] and Vs profile from cross-hole in the Casaglia bore-hole [19,22], (**b**) H/V spectral ratio in the Casaglia site [23,28].

Moreover, ambient vibrations in array mode were recorded for the seismic microzonation aimed at supporting the Emilia 2012 post earthquake reconstruction. The tests carried out along the Mirandola buried ridge gave results consistent with the cross-hole test: the Vs profile from the Mirandola array shows an important Vs contrast at depth of about 110 m, below which Vs > 800 m/s (compares Figures 15 and 16; see also Figures S2–S5 in the Supplementary Materials).

### 4.2.2. Other Areas

In the Emilia-Romagna plain, direct measurements of Vs at depths greater than 50 m are almost non-existent (except for the high structural zones). Therefore, data for the identification of the seismic bedrock mainly derive from ambient noise measurements and comparisons with the geological cross-sections.

In the syncline and on the limbs, different tests for the estimation of Vs at high depths were carried out near Mirabello and in Cavezzo.

Minarelli et al. (2016) [36] performed Vs measurement with a down-hole test in a 300 m deep borehole, just south of Mirabello. The alluvial sediment base was recognised at a depth of about 290 m, but the Vs measurement was carried out up to a depth of 265 m;

therefore, direct Vs measurements in the geological substratum are not available. In any case, this down-hole provides important information on Vs in alluvial sediments. The records did not show important Vs discontinuities (Figure 18a): Vs gradually increases for the first 100 m, up to the AES6 base, and in the lower part of AES it is generally about 400 m/s. A Vs increase is observed at about 160 m, passing from AES to the underlying sediments (AEI): below this depth, the Vs is generally between 400 and 500 m/s; the maximum value, Vs=525 m/s, was measured at about 260 m.

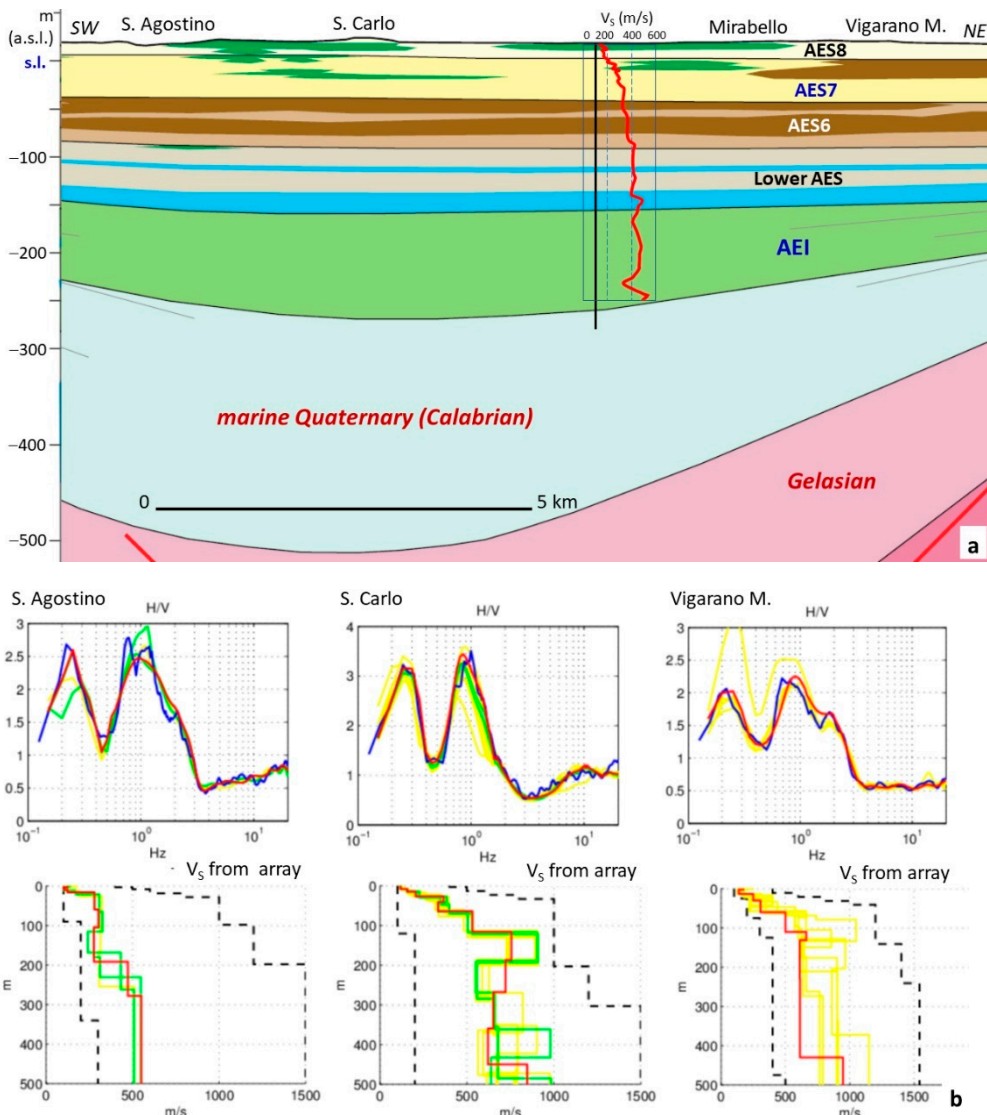

**Figure 18. (a)** Geological cross-section from Sant'Agostino to Vigarano Mainarda [19] and Vs profile from Mirabello down-hole [36], (**b**) H/V spectral ratios and Vs profiles from ambient vibration measurements (array) [19].

Ambient noise measurements (array mode), carried out just south and north of the borehole, are substantially consistent with the down-hole data (Figure 18b). These tests show peaks of the H/V spectral ratios at frequencies of 0.2–0.3 Hz and 0.8–1 Hz; the reconstructed Vs profiles indicate contrasts at depths between 180 and 230 m in Sant'Agostino and between about 100 and 120 m in San Carlo and Vigarano Mainarda. In these last two sites, Vs contrasts between 400 and 450 m were also detected, with Vs > 800 m/s at depths greater than 400 m. Minor Vs contrasts are present at depths of a few tens of meters and about 60–80 m.

Comparing these results on a geological cross-section (Figure 18), we can see that the Vs contrasts can be associated with the basal unconformities of the main stratigraphic units (see also [37]).

In the town of Cavezzo geophysical surveys were carried out by various authors (Figure 19); first for studies in support of post-earthquake reconstruction [19], then for a seismic microzonation pilot study, which was part of the LiquefAct project [26]. The first measurement consists of ambient noise recording (array mode) in the south-east urban area. The result indicates two peaks of the H/V spectral ratio, at frequencies of 0.2 Hz and 0.8 Hz; the reconstructed Vs profile shows two contrasts, the first at a depth of about 80–90 m and the second at a depth of about 240–260 m. Vs reaches 800 m/s below 250 m. Following investigations consist of new ambient noise recordings (array mode), made by INGV in the west side of the urban area, and a high-resolution reflection seismic line, made by OGS just southwest of the INGV array. The Vs profiles reconstructed by OGS and INGV, in collaboration with EUCENTRE, show similar Vs values up to a depth of 170 m and indicate that up to this depth Vs $\leq$ 500 m/s. Below this depth, the Vs profile proposed by INGV-EUCENTRE indicates Vs = 800 m/s while the Vs profile reconstructed by OGS indicates Vs = 500 m/s up to a depth of about 245 m, below which Vs > 700 m/s (Figure 19). The OGS profile is therefore consistent with the Vs profile resulting from the first investigation.

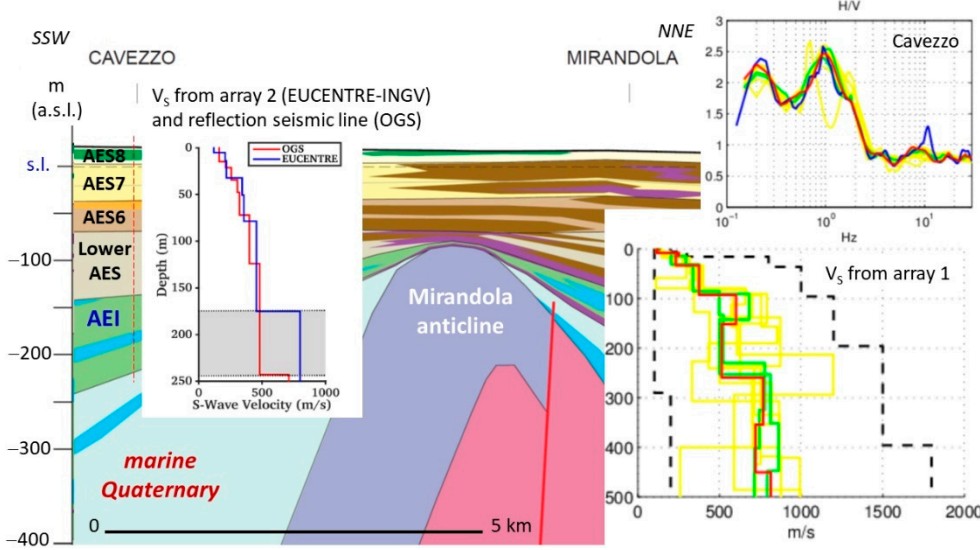

**Figure 19.** Geological cross-section from Cavezzo to Mirandola, H/V spectral ratio and Vs profile from ambient vibration measurement (array) in the Cavezzo urban area [19] and Vs profiles from Cavezzo seismic microzonation [26].

Other examples are available in Supplementary Materials (Figures S6–S10).

A specific discussion should be reserved for the coast north of Rimini, from Cesenatico to the Po delta. This sector is characterised everywhere by poorly consolidated sediments for at least 300 m and the AEI base is, almost throughout the whole sector, at depths greater than 450 m (Figures 5 and 20; see also Figures S11–S14 in Supplementary Materials).

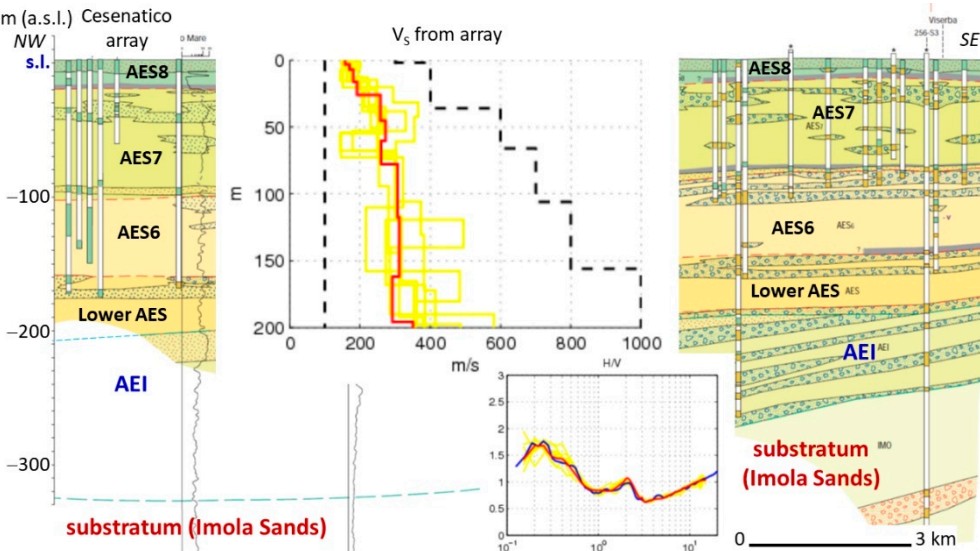

**Figure 20.** Geological cross-section along the coast from Cesenatico to Viserba (from Geological Map of Italy, sheet no. 256 "Rimini"), H/V spectral ratio and Vs profile from ambient vibration measurement (array) in the Cesenatico urban area [38].

Ambient noise measurements, both in single station and in array mode, do not show peaks of the H/V spectral ratio in the frequency range between 0.2 and 10 Hz (Figure 20) and, therefore, important Vs discontinuities were not detected in the first 300 m.

## 5. Results

For the determination of the seismic bedrock, it is also important to consider recent tectonics and how this affected the stratigraphic evolution of the basin.

The Po Plain is an active foredeep basin; so, sedimentation and post-sedimentary evolution of the alluvial units have been affected by the tectonic deformation. The main stratigraphic units, AEI and AES, and related subunits, are in fact bounded by discontinuity surfaces, the oldest of which also show evidence of angular unconformity (Figure 5, Figure 8, and Figure 12; see also Figure S1 in the Supplementary Materials). The maps of these surfaces [11,16] highlight important tectonic deformation at least up to the AES6 (Bazzano subsynthem, 0.230–0.125 Ma). This can explain the differences in the densification of the sediments and the impedance contrasts observed at the boundary surfaces of the main alluvial units.

The case studies illustrated in the previous chapters suggest the following considerations.

The subsoil of the foothills is characterised by plurimetric and decametric horizons of coarse deposits (gravels, sandy and silty gravels) of alluvial fan (Figures 8–10; see also Figure S1 in the Supplementary Materials). The roof of the Upper Pleistocene coarse horizon often constitutes a Vs contrast surface and the underlying gravels often have Vs higher than that of the geological substratum, sometimes reaching and exceeding 800 m/s. In general, it is difficult to identify the seismic bedrock in the foothill areas, but from the comparison between geophysical data and geological cross-sections, it emerges that, in the proximal areas, the seismic bedrock commonly consists of a coarse horizon belonging to the alluvial sequence, at depths of a few tens of meters (Figures 8, 9, 11 and 12), while in the external areas the seismic bedrock is located at depths greater than 100 m and can be associated with the lower horizons of AES or AEI (Figures 13 and 14). In these contexts, the seismic bedrock is not as deep as the geological substratum.

In the distal areas, where the alluvial sequence consists of fine and poorly consolidated sediments (alternating sands, silts, and clays), direct measurements indicate that the Vs is never high (Vs < 500 m/s) and reaches 800 m/s only in correspondence of the geological substratum (Figures 15 and 18b).

At the top of the main buried structural highs, where the thickness of alluvial sediments is less than 150 m, the ambient noise measurements indicate a single and clear Vs contrast. In these environments, the seismic bedrock is generally easily identifiable and coincides with the geological substratum (Figures 15–17).

In the other areas of the Emilia-Romagna plain, where the thickness of alluvial sediments is greater than 150 m (regardless of the geometry of the geological substratum), the geophysical measurements usually show two Vs contrasts (Figures 18 and 19; see also Figures S6–S8 in the Supplementary Materials): the shallower one is always associated with a basal unconformity of an alluvial unit, while the second is often associated with the roof of the geological substratum or at greater depths. In these contexts, for the correct calculation of the expected shaking on the surface, the seismic bedrock can be ascribed to the geological substratum, but the presence of important Vs contrasts in the overlying succession must always be evaluated.

In the Po delta and along the Adriatic coast, where the thickness of alluvial sediments is greater than 300 m everywhere, geophysical measurements indicate the absence of significant velocity contrasts, and the Vs seems to gradually increase with depth. Vs values tend to be low in the upper alluvial cycle (AES, see Figure 20; see also Figures S11–S13 in the Supplementary Materials). The seismic bedrock is at a depth of at least 300–350 m and coincides with the first alluvial cycle (AEI) or with the geological substratum.

Table 2 summarizes the main geological and geophysical characteristics recognized in the Emilia-Romagna plain and described above.

**Table 2.** Main geological and geophysical characteristics of the Emilia-Romagna plain.

| Main Characteristics of the Zones | Marginal Area | Distal Areas | | |
|---|---|---|---|---|
| | | | Other Areas | |
| | | Top of the Main Structural Highs | Synclines, Limbs, and Minor Anticlines | Coast and Po Delta |
| **Geological features** | Alluvial sequence made up of plurimetric and decametric coarse horizons | Thickness of the alluvial sequence H ≤ 150 m | Thickness of the alluvial sequence H > 150 m | Thickness of the alluvial sequence H > 300 m |
| **Ground Frequencies** | Fn > 5 Hz | Fn = 0.8–1.4 Hz | Fn = 0.2–0.3 Hz; Fn = 0.7–Hz | No Fn in the 0.2–10 Hz range |
| **Seismo-Stratigraphic Discontinuities** | Top of the shallower plurimetric coarse horizon (usually Upper Pleistocene age) | Top of the geological substratum | Basal unconformities of the main alluvial units (e.g., AES6, AES, AEI) | No seismo-stratigraphic discontinuities |
| **Seismic Bedrock** ($Vs_{bedrock}/Vs_{cover} > 2$ or $Vs ≥ 800$ m/s) | Plurimetric/decametric coarse horizon (Middle Pleistocene age), usually at depth between 40 m and 100 m | Geological substratum | Lower alluvial cycle (AEI) or geological substratum | Lower alluvial cycle (AEI) or geological substratum |

The data presented are in good agreement with the ground frequencies map and the seismic bedrock map proposed by Mascandola et al. (2019) [3]. The comparison between the data presented and Figure 9 by Mascandola et al. (2019) highlights that the depth indicated by these authors as the roof of the seismic bedrock certainly corresponds to the depth of the main impedance contrast observed but does not always correspond to the roof of rocks characterised by $Vs ≥ 800$ m/s ("engineering bedrock"). In any case, the map proposed by Mascandola et al. (2019) remains the best reference for the estimation of the main Vs discontinuity in the Po Plain.

## 6. Conclusions

Where the thickness of unconsolidated sediments is high and it is not possible to measure the Vs with direct investigations up to the seismic bedrock, the least expensive way to identify the latter is by comparing the results of the ambient noise measurements

(preferably in array), the derived Vs profiles, and the available lithologic and stratigraphic data (borehole logs and geological cross-sections).

This approach was tested in the Emilia-Romagna plain and produced the following results (see also Table 2):

- In the foothills, where the coarse deposits of alluvial fan are prevalent, the seismic bedrock is sometimes difficult to identify; the roof of the first coarse horizon from the surface often constitutes an important velocity discontinuity and Vs already exceeds 800 m/s in the gravel horizons at a depth of a few tens of meters;
- In the distal areas, where the alluvial sequence is made up of fine sediments (alternating sands, silts, and clays), the Vs of alluvial deposits is generally less than 500 m/s; in this context, three typical situations were recognised:
  1. At the top of the higher buried ridges, where the thickness of alluvial sediments is less than 150 m, the seismic bedrock is generally well recognizable and tends to coincide with the geological substratum;
  2. In the other areas of the Emilia-Romagna plain, the thickness of alluvial sediments is greater than 150 m and the geophysical data usually indicate two important Vs contrasts: the first generally corresponds to the basal unconformity of one of the alluvial units (AES6 or AES), while the second generally corresponds to the base of alluvial deposits (or the roof of the geological substratum); in these successions the seismic bedrock seems to correspond to the lower alluvial cycle (AEI), but it cannot be excluded that it is deeper and coincides with the geological substratum;
  3. Along the Adriatic Coast and in the Po delta the thickness of alluvial sediments is greater than 300 m everywhere and the geophysical data indicate the absence of significant velocity contrasts; Vs is generally low in the whole upper alluvial cycle and reaches 800 m/s at depth of at least 300–350 m; in this context as well, the seismic bedrock seems to correspond to the lower alluvial cycle (AEI) or to the geological substratum.

Finally, based on the presented data, the map of the seismic bedrock depth proposed by Mascandola et al. (2019) does not always seem to indicate the roof of the "engineering bedrock" (Vs $\geq$ 800 m/s), but rather the depth of the main Vs contrast surface. In spite of that, the work of Mascandola et al. (2019) remains a fundamental reference for site seismic assessment in the Po Plain.

**Supplementary Materials:** The following are available online at https://www.mdpi.com/article/10.3390/geosciences11070297/s1, Figure S1: Marginal area: comparison between stratigraphic log, Vs-Vp profiles and ambient noise measurements in Viserba (Rimini Municipality) [32,33]. Figure S2: Distal areas, high structural zones: comparison between stratigraphic data [11], H/V spectral ratio and Vs profile from ambient vibration measurement (array) in the Novi di Modena urban area [19], on the westward continuation of the blind Mirandola anticline. Figure S3: Distal areas, high structural zones: comparison between stratigraphic data, H/V spectral ratio and Vs profile from ambient vibration measurement (array) in the San Felice sul Panaro urban area [19], on the eastward continuation of the Mirandola anticline. Figure S4: Distal areas, high structural zones: H/V spectral ratios and Vs profiles from ambient vibration measurement (array) around the Mirandola anticline area [19]. Figure S5: Distal areas, high structural zones: H/V spectral ratios and Vs profiles from ambient vibration measurements (array) in Bondeno and Scortichino, on the intermediate structural high of the Ferrara Folds [19]. Figure S6: Distal areas, other areas: comparison between stratigraphic data (geological cross-section from Geological Map of Italy, sheet no. 202 "S. Giovanni in Persiceto"), H/V spectral ratio and Vs profile from ambient vibration measurement (array) in the Crevalcore urban area [19]. Figure S7: Distal areas, other areas: comparison between stratigraphic data, H/V spectral ratio and Vs profile from ambient vibration measurement (array) in the Cento urban area [19]. Figure S8: Distal areas, other areas: comparison between stratigraphic data (geological cross-section from Geological Map of Italy, sheet no. 203 "Poggio Renatico"), H/V spectral ratio and Vs profile from ambient vibration measurement (array) in the Poggio Renatico urban area [19]. Figure S9: Distal areas, other areas: H/V spectral ratios and Vs profiles from ambient vibration measurements

(array) in Reggiolo and Camposanto urban areas [19]. Figure S10: Distal areas, other areas: comparison between stratigraphic data, H/V spectral ratios and Vs profiles from ambient vibration measurements in the Boretto area [28]. Figure S11: Distal areas, coast and Po delta: comparison between stratigraphic data (geological cross-section from Geological Map of Italy, sheet no. 187 "Codigoro"), H/V spectral ratios and Vs profiles from ambient vibration measurements (array) in the Codigoro and Goro urban areas [38]. Figure S12: Distal areas, coast and Po delta: comparison between stratigraphic data (geological cross-section from Geological Map of Italy, sheet no. 205 "Comacchio"), H/V spectral ratio and Vs profile from ambient vibration measurement (array) in the Comacchio urban area [38]. Figure S13: Distal areas, coast and Po delta: comparison between stratigraphic data [11], H/V spectral ratio and Vs profile from ambient vibration measurement (array) in the Casal Borsetti area [38]. Figure S14: Distal areas, coast and Po delta: H/V spectral ratios and Vs profiles from ambient vibration measurements (array) along the coast from Porto Garibaldi to Cervia [38].

**Funding:** This research received no external funding.

**Institutional Review Board Statement:** "Not applicable" for studies not involving humans or animals.

**Informed Consent Statement:** "Not applicable" for studies not involving humans.

**Data Availability Statement:** The data presented in this study are available within the article or supplementary material.

**Conflicts of Interest:** The authors declare no conflict of interest.

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
