# Peer review of "Assessment of Seismic Bedrock in Deep Alluvial Plains. Case Studies from the Emilia-Romagna Plain"

_geosciences, doi:10.3390/geosciences11070297_

Round 1

Reviewer 1 Report

In the introduction there are bibliographical references relating to the proposed study and figure 1 is useless. There is talk of data taken from the literature but they are not cited. Table 1 does not specify where and how the illustrated data were obtained.

There are no references to very important previous studies and above all to previous studies on the relationship between basement mapping with single station and array ambient vibration data (e.g. Arai & Tokimatsu, 2005; Bonnefay-Claudet et alii, 2006 and 2008; Milana et alii, 2014; Luzi et alii, 2013; Castellaro & Mulargia, 2016; Albarello & Mucciarelli; Massa et alii, 2016; Mascandola et alii, 2017 and 2019);

The chapter relating to the "Geological framework" is completely devoid of previous studies that investigated the Po valley and described in detail the filling sediments of the plain and the characteristic depositional facies (eg AGIP, 1978; Bruno et alii, 2015; Campo et alii, 2016; Amorosi et alii, 2008 and 2017)
There are no significant references on the structural set-up and geological and geomorphological evolution of the Po valley (eg Capozzi & Castellarin, 1992; Barozzi & Colombi, 1992; CROP 01; Farabegoli et alii, 1997; Castiglioni, 1999).

In general, the figures are not very legible and in some cases the legend is not complete and / or clear. It is necessary to improve the graphics of the pictures and the captions. I recommend changing the definition of quality with a higher contrast color.

Author Response

Figures: Figure 1 deleted as suggested; increased size and contrast of text fonts.

About the other requests:

It is important to consider that the purpose of this paper is not a review or discussion on the stratigraphic and geomorphological evolution of the Po Plain or on the use of ambient vibrations for the characterization of the subsoil. The main goal is the proposal to use current (and low-cost) geophysical tests associated with lithostratigraphic information for the recognition of seismic bedrock. Unfortunately, the recognition of the seismic bedrock is still too often not sufficiently investigated in seismic microzonation and in seismic site assessment, especially in urban areas with high thickness of soft sediments.

For this reason, I have not cited the numerous works on the use of environmental vibrations for the characterization of the subsoil and on the stratigraphy and tectonics of the Po Plain, but I preferred to give more information on where to find lithostratigraphic data.

So, instead of citing the numerous works on the ambient vibrations, I believe it is sufficient refer to the literature reviews by Bonnefay-Claudet et al., 2006.

With regard the geological framework, the first fundamental work that described the geological structure of the Po Plain was the synthesis of subsoil data (bore-hole logs and seismic lines, from AGIP database) published by Pieri & Groppi, 1981. Finally, the stratigraphy of the Po Plain alluvial sequence was described and defined, on a regional scale, in the studies on the Po aquifers carried out by the Emilia-Romagna and Lombardy regions, in collaboration with ENI-AGIP (respectively 1998 and 2002). These are, in my opinion, the reference works for the geological framework of this paper.

Reviewer 2 Report

see 'pencilled' comments and suggestions in the attached file

Author Response

All corrections indicated by Reviewer 2 were accepted

Reviewer 3 Report

The paper is not prepared upon „Instructions for Authors/Manuscript Preparation“ (https://www.mdpi.com/journal/geosciences/instructions#preparation). Actually, the text is sloppy written, numerous corrections are needed in order to uniformize and harmonize with the standards of the journal itself. Extensive English proofreading is also required.

Some of the comments are inserted into the attached PDF file. The other remarks are:
- variables should be written uniformly;
- a lot of data sources are missing;
- Figure resolution should be improved; in most cases font size should be enlarged; most figure captions need to be supplemented with explanation of data displayed on figures; the locations of boreholes and cross-sections must be indicated on one of the maps; the coordinates should be inserted on the maps.

But, my major remark is:
The paper is basically not scientific, it combines results from various professional studies, and in fact has no scientific contribution. It is a professional study that combines all available measurements/results for certain area in Italy. It is also too long to be able to read and understand well. It is almost impossible to connect the structural interpretations in figures with the geographical position of the mentioned structures, so the basic meaning of the work itself is lost. It would be necessary to present the results in a better/condensed manner, to be focused to representative ones (and to put the others, e.g., in the supplement). Also, to emphasize the aim, importance and scientific contribution of the paper itself, taking into account that the article is intended to be read by scientists outside Italy. In this regard should be limited to mentioning the most important geographical areas, which must be clearly indicated in the figures.

Author Response

The manuscript has been corrected considering the Instructions for the Authors and framing text and figures in the template. The revisions requested in your attached file were all accepted. The font size and contrast of the text in the figures have been improved and all data sources have been included. The locations of the tests and geological cross-sections are indicated in figure 4.

I do not think it useful to include more details in the captions of the figures; this would lengthen the text without providing more information. I believe that the figures are sufficiently described and commented on in the text. References to figures in the text are clearly indicated.

As for the major remark, in my opinion, the non-scientific nature of the work is not a negative aspect. The paper is in fact proposed for a special issue whose topic is "Seismic Microzonation Analysis of the Anthropized Environment: Approaches and New Perspectives" and originates from my daily experience (I work for a Geological Survey of a Regional Authority). Unfortunately, despite the many scientific works on the subject, there are still numerous seismic microzonation studies for urban planning and seismic assessments for building design that do not carefully evaluate the depth of the seismic bedrock, because of the difficulty of the investigation, producing unrealistic results. Therefore, for the special issue, I believe that a technical contribution has a value no less than that of a scientific contribution.

Finally, I believe that many examples are more significant than a selection. One of the purposes of the paper is to show the reproducibility of the proposed procedure and to illustrate the large amount of data available.

Round 2

Reviewer 1 Report

Bibliographic citations are not always consistent with the considerations, data and figures presented (figure 3 and citation 16).

In the abstract the author talks about 4 domains while in the cap.4 he talks about 3 domains.

The author does not explain why he does not utilize the subdivision published by other authors to explain the filling of the Po valley with respect to his proposal.

Author Response

Request 1: Bibliographic citations are not always consistent with the considerations, data and figures presented (figure 3 and citation 16). Reply: the citation 16 in Figure 3 is correct; the geological cross-section in question was created for the Seismotectonic Map of the Emilia-Romagna Region (2016 edition) and published here for the first time

Request 2: In the abstract the author talks about 4 domains while in the cap. 4 he talks about 3 domains. Reply: corrected the sentence at the beginning of chapter 4 (line 196)

Request 3: The author does not explain why he does not utilize the subdivision published by other authors to explain the filling of the Po valley with respect to his proposal. Reply: In my opinion, the request is not clear; in any case, I am not aware of previous subdivisions of the Po Plain, or Emilia-Romagna plain, relating to the topic of the article. Therefore, no changes have been made in the text in this regard

Reviewer 3 Report

The paper has been significantly improved in terms of requests for its preparation by the journal. However, nothing else has been done and it is still a (too long) technical work (and useful for professional purpose), which has no elements of scientific and as such I am of the opinion that it is not suitable for publication in a respectable journal such as Geosciences.

Author Response

Reply: I completely disagree with the Reviewer 3's comment.

I know that the text is more technical than scientific but it must be considered that the article has been proposed for a special issue whose topic is seismic microzonation in anthropized areas, experiences and proposals for approaches. I believe that the decision on the relevance of the text and the opportunity to publish it in the special issue must be made by the Special Issue Editors.

Finally, I consider the large number of examples an added value, which demonstrates the applicability of the proposed procedure and the availability of numerous geological data, unfortunately too often ignored.